# The Association between Gut Microbiome and Pregnancy-Induced Hypertension: A Nested Case–Control Study

**DOI:** 10.3390/nu14214582

**Published:** 2022-11-01

**Authors:** Huijun Lin, Junru Chen, Shujuan Ma, Rongjing An, Xingli Li, Hongzhuan Tan

**Affiliations:** 1Hunan Provincial Key Laboratory of Clinical Epidemiology, Xiangya School of Public Health, Central South University, Changsha 410000, China; 2Department of Epidemiology and Health Statistics, Xiangya School of Public Health, Central South University, Changsha 410000, China; 3Clinical Research Center for Reproduction and Genetics in Hunan Province, Reproductive and Genetic Hospital of CITIC-Xiangya, Changsha 410000, China

**Keywords:** pregnancy-induced hypertension, gut microbiome, metagenomic sequencing, nested case–control study

## Abstract

(1) Background: Pregnancy-induced hypertension (PIH) is associated with obvious microbiota dysbiosis in the third trimester of pregnancy. However, the mechanisms behind these changes remain unknown. Therefore, this study aimed to explore the relationship between the gut microbiome in early pregnancy and PIH occurrence. (2) Methods: A nested case–control study design was used based on the follow-up cohort. Thirty-five PIH patients and thirty-five matched healthy pregnant women were selected as controls. The gut microbiome profiles were assessed in the first trimester using metagenomic sequencing. (3) Results: Diversity analyses showed that microbiota diversity was altered in early pregnancy. At the species level, eight bacterial species were enriched in healthy controls: *Alistipes putredinis*, *Bacteroides vulgatus*, *Ruminococcus torques*, *Oscillibacter unclassified*, *Akkermansia muciniphila*, *Clostridium citroniae*, *Parasutterella excrementihominis* and *Burkholderiales bacterium_1_1_47*. Conversely, *Eubacterium rectale*, and *Ruminococcus bromii* were enriched in PIH patients. The results of functional analysis showed that the changes in these different microorganisms may affect the blood pressure of pregnant women by affecting the metabolism of vitamin K_2_, sphingolipid, lipid acid and glycine. (4) Conclusion: Microbiota dysbiosis in PIH patients begins in the first trimester of pregnancy, and this may be associated with the occurrence of PIH. Bacterial pathway analyses suggest that the gut microbiome might lead to the development of PIH through the alterations of function modules.

## 1. Introduction

Pregnancy-induced hypertension (PIH) is defined as an elevated blood pressure that occurs for the first times during pregnancy, and gestational hypertension (GH) and pre-eclampsia (PE) are the two most common types [1,2]. The global incidence of PIH is 4.1–19.4%, with GH accounting for 1.8–4.4% of cases and PE accounting for 0.2–9.2% of cases [3]. Without the implementation of prevention measures, 25–50% of GH patients will develop PE [4]. PIH can adversely affect pregnant women and their fetuses [5,6]. Studies have found that PIH is the most common reason for death in pregnant women in both Asian and European countries [7,8]. Most PIH patients will recover a normal blood pressure after delivery. However, nowadays, more and more studies have discovered that women with a history of PIH have a higher risk of hypertension, stroke, and ischemic heart disease in future life [9,10]. Furthermore, the offspring of mothers with PIH present with higher blood pressure and increased risk of other diseases than children of healthy mothers [11,12,13,14,15]. Therefore, with the increasing incidence of PIH and the associated serious health damage, PIH is considered an urgent public health problem. However, the specific pathogenesis of PIH remains uncertain.

The gut microbiome is the human body’s biggest and most complex biocenosis [16]. The number of genes in the gut flora is considerably more than that in the human genome, and the gut microbiome composition varies greatly in individuals because of genetic and environmental variations [17]. The gut microbiome maintains a dynamic symbiotic relationship through interaction with the immune system. It can produce metabolites to inhibit the excessive colonization of harmful bacteria, thereby participating in the regulation of human health [18]. Many studies have found a close relationship between the gut microbiome and diseases, such as obesity, diabetes, and cardiovascular diseases [19,20].

To provide adequate nutrition to the fetus during pregnancy, maternal physiology undergoes adaptive reconstruction. At this time, the gut microbiome is more sensitive to external stimuli, and dysbiosis is prone to occur, leading to the development of pregnancy complications, including gestational diabetes and hyperemesis gravidarum [21,22]. Studies have shown that the gut microbiome is associated with blood pressure, specifically regulating the blood pressure when the body is in low-grade chronic inflammation [23]. Moreover, many microbial metabolites are involved in the regulation of blood pressure [24,25,26].

Ley et al. found that the abundance of *Fusobacterium* and *Veillonella* was enriched, and the abundance of *Faecalibacterium* and *Akkermansia* was decreased in PE women [27]. In obese pregnant women, the abundance of butyrate producing bacteria, such as *Odoribacter* and *Clostridiaceae*, was negatively correlated with systolic blood pressure (SBP) [28]. The gut microbiome in late pregnant PE patients had lower diversity and obvious dysbiosis, and animal experiments have shown that dysbacteriosis may cause high blood pressure [29]. Moreover, it has been reported that the dysbiosis continues during postpartum [30].

Microbiota dysbiosis has been found in PIH women in the third trimester. However, the changes in the gut microbiome in PIH women in early pregnancy remain unknown. Furthermore, the potential mechanism of the relationship between the gut microbiome and PIH has not been explored. Therefore, in this study, we conducted a nested case–control study to evaluate the alteration of microbiota taxa and function modules in the first trimester and detect the correlation between microbiota taxa, function modules, and clinical indices to reveal the mechanisms of the gut microbiome and PIH.

## 2. Materials and Methods

### 2.1. Study Design

This nested case–control study was based on an early pregnancy follow-up cohort. The cohort was established in the Hunan Provincial Maternal and Child Health Hospital (HPMCHH) from March 2017 to March 2018. The project was approved by the Medical Ethical Committee of HPMCHH (NO: EC201624) on 11 January 2017. All eligible subjects provided written informed consent and then entered the cohort.

The inclusion criteria were: single embryo; natural conception; no history of diabetes, hypertension or other diseases before pregnancy; no acute infection in the past 2 weeks; no antibiotic use during pregnancy. Participants were recruited in the first trimester (10–14 weeks) and followed up until 42 days postpartum. The collection of questionnaire results and samples were conducted in the first trimester, second trimester, third trimester, at delivery, and during postpartum. In addition, biochemical tests of all blood samples were performed at the hospital.

Women in the cohort who were diagnosed with PIH after 20 weeks gestation by the hospital, and had complete questionnaire information, stool samples, and blood samples, were selected as a case group. The same number of healthy pregnant women were selected as a control group matched for age (±3 years), gestational age (±1 week) and stool/blood sample collection time (±1 month). All biological samples were collected in the first trimester.

### 2.2. DNA Extraction and Metagenomic Sequencing

Fecal DNA was extracted using QIAamp Fast DNA Stool Mini Kit (Qiagen, Hilden, Germany). All samples were paired-end sequenced on the Illumina platform (read length 150 bp) at Novogene Bioinformatics Technology Co Ltd. (Tianjin, China). After quality control using Trimmomatic [31] (Version 0.36), reads aligned to the human genome (alignment with SOAP2 [32], Version 2.21) were removed. The remaining high-quality reads were used for further taxonomic annotation and functional annotation by MetaPhlAn [33] (Version 2.1.0) and HUMAnN [34] (Version 2-0.11.0), respectively.

### 2.3. Statistical Analysis

Means ± standard deviations and medians were used for statistical description. Matched t-tests, ANOVA, and nonparametric tests were used for statistical inference. All the statistical analyses were two-tail tests, and *p* < 0.05 indicated statistical significance. Rarefaction analysis was performed to assess the species richness in the controls and cases by vegan package in R software. Alpha diversity indices were estimated using the vegan package (Version 2.5-7) in R software (Version 4.0.2) based on the species profile. A principal coordinates analysis (PCoA) was executed using the ade4 package (Version 5.5) in R software. A non-metric multidimensional scaling (NMDS) analysis was performed using the vegan package in R software. Differential abundance of microorganisms and functional modules was tested using MaAsLin2 [35] (Version 1.2.0). Only features existing in at least 10% of subjects were considered in the analyses. Correlation analyses were tested using Spearman’s correlation with the psych package (Version 2.1.9) in R software.

## 3. Results

### 3.1. Main Results

#### 3.1.1. General Characteristics

A total of 872 subjects were included in the cohort study, and 744 completed entire pregnancy to postpartum investigations. A total of 46 PIH patients (29 with GH and 17 with PE) were diagnosed in the cohort, of which nine GH subjects and two PE subjects were excluded due to lack of questionnaire information or stool and blood samples. Finally, 35 PIH subjects (20 with GH and 15 with PE) were selected in this nested case–control study, and 35 healthy controls were selected according to the match factors. All the results of subgroup, which is PE and GH, and control are showed in Appendix A.

The basic characteristics of the two groups are presented in Table 1. The results show no significant differences between the two groups in age, gestational age, race, occupation, education level, and monthly income. (The basic characteristics of PE and GH group are showed in Appendix A).

The clinical characteristics of the two groups subjects are presented in Table 2. The results show waist, weight, BMI, SBP, diastolic blood pressure (DBP), hemoglobin (HGB), uric acid (UA), insulin (INS), and γ-glutamyltransferase (GGT) in early pregnancy were higher in the PIH group than in controls (*p* < 0.05). (The clinical characteristics of PE and GH group are showed in Appendix A).

#### 3.1.2. The Gut Microbiome Diversity Alteration

The rarefaction curve was performed according to the gut microbiome data from metagenomic sequencing. The curve was flat in our sample size, indicating that sufficient sequencing data were obtained to reflect nearly all microbial diversity in the subjects (Figure 1).

The results show no significant differences in alpha diversity between the two groups. Inversely, the beta diversity between the PIH and control groups did differ statistically according to the PCoA results and NMDS analysis (Figure 2A,B). (The result of diversity analysis between three groups is showed in Appendix A).

#### 3.1.3. Microbial Taxa Alteration

Comparison of the PIH and control groups revealed three, four, four, six, seven and ten different taxa at the phylum, class, order, family, genus and species level, respectively.

At the phylum level, PIH patients had a higher abundance of *Firmicutes* (*p* = 0.025) and a lower abundance of *Bacteroidetes* (*p* = 0.020) and *Verrucomicrobia* (*p* = 0.026) (Figure 3A).

At the species level, PIH patients had significantly higher abundance of *Eubacterium rectale* (*p* = 0.009) and *Ruminococcus bromii* (*p* = 0.039) and a lower abundance of *Alistipes putredinis* (*p* = 0.003), *Bacteroides vulgatus*
*(p* = 0.007), *Ruminococcus torques* (*p* = 0.011), *Oscillibacter unclassified* (*p* = 0.006), *Akkermansia muciniphila* (*p* = 0.026), *Clostridium citroniae* (*p* = 0.039), *Parasutterella excrementihominis* (*p* = 0.043), and *Burkholderiales bacterium_1_1_47* (*p* = 0.047) (Figure 3B). (The taxa alterations between three groups are showed in Appendix A).

#### 3.1.4. Functional Prediction Analysis

The transcriptome information of samples was compared with KO and GO standard libraries to explore the potential function and mechanism of the gut microbiome in the occurrence of PIH. The preliminary screening of different functions was conducted according to “*p* < 0.01”, then through literature review, related functions were chosen and visually presented in a heatmap. As shown in Figure 4, PIH patients had significantly lower levels in five functions. (The functional alteration between three groups is showed in Appendix A).

#### 3.1.5. Correlation Analysis

##### Correlation between the Gut Microbiome and Clinical Factors

The correlation analysis was conducted on clinical factors and the 10 different microbial species screened in the PIH and control groups. Five bacteria were associated with 11 clinical indicators. The abundance of *A. tredinis* was negatively correlated with early UA, early SBP, and early DBP and positively correlated with LDLCH. The abundance of *B. vulgatus* was negatively correlated with early BMI, waist, early ALT, early UA, early weight, early AST, INS, GGT, as well as early SBP and early DBP. The abundance of *O. unclassified* and *A. muciniphila* was negatively correlated with early SBP and early DBP, and *A. muciniphila* was positively correlated with GGT. The abundance of *C. citroniae* was negatively correlated with early DBP (Figure 5). (The correlation between gut microbiome and clinical factors of three groups is presented in Appendix A).

##### Correlation between Gut Microbiome and Functions

The results of the association analysis between different microbial species and functions indicated that *A. putredinis*, *B. vulgatus* and *O. unclassified* were positively associated with the five functions (Figure 6). (The correlation between gut microbiome and functions of three groups was presented in Appendix A).

## 4. Discussion

In this study, we explored the differences in gut microbiome between PIH patients and healthy controls in the first trimester of pregnancy. The results show that the gut microbiome is changed in early pregnancy, and the changes might be associated with the occurrence of PIH. Moreover, all the subjects did not take drugs two weeks before enrollment, and they are all permanent residents in Hunan, with no obvious difference in diet. The influence of external factors in the gut microbiome was excluded as far as possible.

### 4.1. Diversity Alteration

The gut microbiome is interdependent with the host in normal physiology, and this interdependency maintains dynamic balance. When microbiota diversity is reduced, a variety of diseases might occur [36,37]. Our study showed no significant differences in alpha diversity but greatly significant changes in beta diversity. Similarly, Wang et al. [38] obtained the same result in PE patients. However, another study found that alpha and beta diversity in PE patients were both significantly reduced [29]. This indicates that the bacterial proportions change in the first trimester of pregnancy, suggesting that the occurrence of PIH might be related to specific bacteria.

### 4.2. Microbial Taxa Alteration

Most significantly different strains were mainly focused on *Firmicutes* and *Bacteroidetes* in our study. It was been reported in 2019 that PE patients have a decreased abundance of *Firmicutes* and an enriched abundance of *Bacteroidetes* [38], which is contrary to our study’s results. On the contrary, another study found that the abundance of *Firmicutes* and *Bacteroidetes* was not significantly different in PE patients and healthy pregnant women in the second trimester, however, the opposite difference was observed in the third trimester [39]. This indicates that the gut microbiome is in flux during disease and gestation and opposing patterns may be present depending on the sampling time and study population.

*E*. *rectale* is an opportunistic pathogen in PIH-enriched species, which can easily cause a series of health problems. Significant enrichment of this species has been found in hypertensive patients [40], indicating that it could increase the host’s blood pressure.

In PIH-reduced species, *C*. *citroniae*, together with *Oscillibacter*, has been reported to be negatively associated with visceral obesity and metabolic syndrome, and it is significantly enriched in healthy people [41,42]. Moreover, studies found that PE patients had an enriched abundance of *Clostridium* [38]. Although the role of *B*. *vulgatus* and *A*. *putredinis* in PIH has not yet been reported, animal experiments have found that *B*. *vulgatus* can inhibit inflammatory processes and enhance the intestinal barrier [43]. Furthermore, reports suggest that *Alistipes* may reduce inflammation [44], and that it may have protective effects on colitis, immune diseases, and cardiovascular diseases, and that the abundance of *Alistipes* is decreased in PE patients [38,45]. The decreased abundance indicates that PIH patients might already have an obvious inflammation in the first trimester, and long-term low-grade inflammation during pregnancy has been proven to induce elevated blood pressure.

In recent years, *A*. *muciniphila* has been regarded as a potentially beneficial bacterium [46], closely associated with metabolic and gastrointestinal diseases [47,48]. It has been reported that the abundance of *Akkermansia* is significantly decreased in PE women [30]. The decreased abundance of *A*. *muciniphila* in PIH women could increase intestinal permeability and the probability of harmful intestinal metabolites entering the bloodstream. *R*. *torques* can damage the intestinal barrier and cause inflammation [49], which is contrary to our study’s results. The mechanism of such reverse alteration remains unclear, and this may be related to the compensatory increase in the beneficial bacteria in the early stages of PIH.

### 4.3. Exploration of the Potential Mechanisms of the Gut Microbiome

We found that the menaquinone biosynthetic process module was lower in PIH patients. Menaquinone is a generic term for vitamin K_2_, which is synthesized by the gut microbiome. Studies have found that vitamin K_2_ is a protective factor against cardiovascular and metabolic disease [50]. Its deficiency might cause vasculopathy and increase in blood pressure. Other studies have suggested a reduced vitamin K_2_ level could be associated with insulin resistance and damage to the kidney and liver [51,52,53]. On the other hand, PIH patients had a significantly higher level of UA, GGT and INS in our study, which indicated the possibility of kidney and liver damage and metabolic abnormalities, and these alterations were found to be associated with PIH [54,55,56,57]. It is consistent that the changes of these indicators and functional modules, suggesting that they might interact with each other to participate in the occurrence of PIH.

The lower sphingolipid metabolic process module in PIH patients meant a higher level of sphingolipids. Studies suggest that ceramide, one of the most common sphingolipids, is related to the angiotensin-aldosterone system and could elevate blood pressure [58]. In addition, ceramide accumulation could promote liver steatosis and cause liver damage [59]. Studies have found that a higher ceramide level in obese people enhances insulin resistance [60,61], and liver injury and insulin metabolism disorders are both related to PIH. It is suggested that lower sphingolipid metabolism in PIH patients might be involved in the occurrence of PIH through the interaction with the level of blood pressure and insulin.

The lipoyl synthase (LipA) module was lower in PIH patients, and LipA is a key enzyme in lipoic acid synthesis. The lower synthesis of LipA in PIH patients suggests lower lipoic acid levels. Lipoic acid could relieve oxidative stress and insulin resistance [62], and could reduce blood pressure in hypertension [63]. Lower lipoic acid levels, higher oxidative and insulin level in the first trimester might work together to form the basics of PIH.

The glycine cleavage system (GCS) is the main pathway of glycine metabolism. Our study found that the aminomethyltransferase and glycine cleavage system H protein, which were components of GCS, were lower in PIH women. The decreased GCS contents suggest higher glycine in PIH patients. Glycine is an amino acid with anti-inflammatory effects [64]. The increased content in PIH patients indicates that some physiological adjustments have been made in these patients in early pregnancy in response to the pathological changes before PIH occurred. This might be related to compensations mechanisms.

Taken together, these results suggest that differences in the gut microbiome might affect the metabolic functional modules, which interact with physiological function alterations, ultimately the PIH happened.

The main limitation of our study was the small sample size which may reduce the statistical power of our results. Moreover, this study did not explore the function of the inflammatory factors and beneficial microbial metabolites involved in the association between the gut microbiome and PIH. Further studies are needed to explore the mechanisms of the gut microbiome in PIH.

## 5. Conclusions

This is the first study to investigate the association between gut microbiome alterations in the first trimester and the occurrence of PIH. The results show that alterations in the microbiome begin in the first trimester, including alterations in the microbial diversity, taxa, and function. These differences in the gut microbiome affect the metabolic functional modules and clinical indicators, thereby promoting PIH occurrence.

## Figures and Tables

**Figure 1 nutrients-14-04582-f001:**
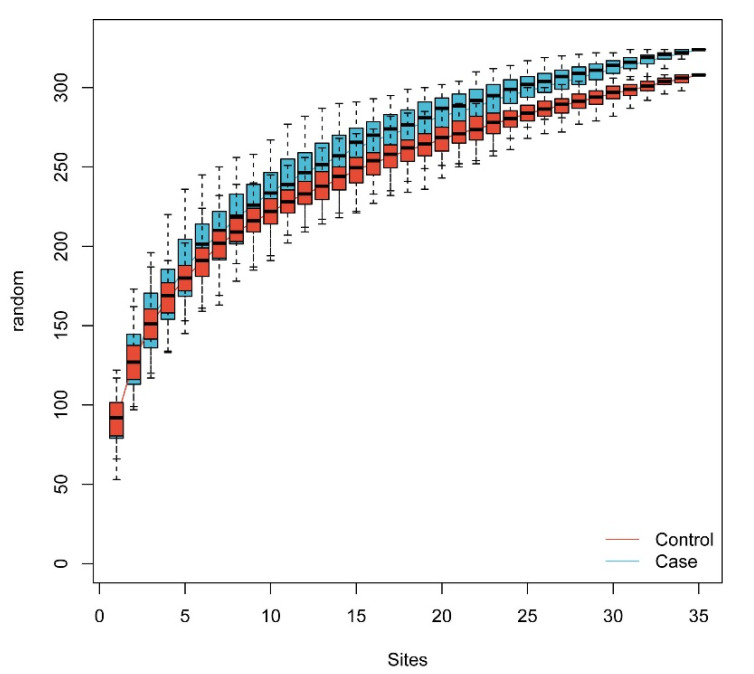
Rarefaction curve.

**Figure 2 nutrients-14-04582-f002:**
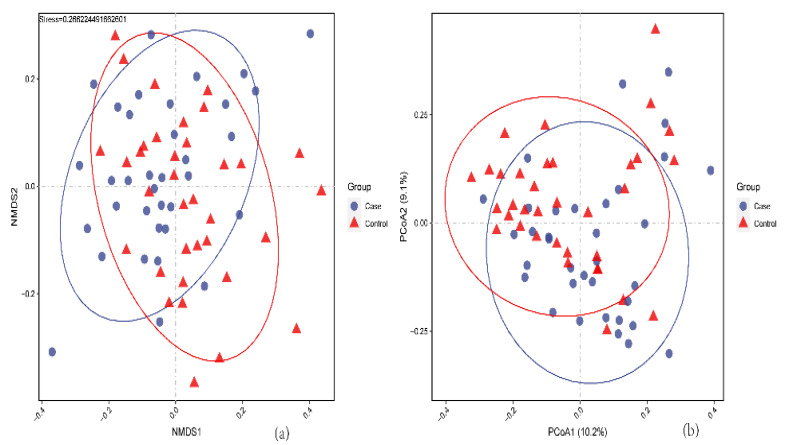
The beta diversity of PIH and healthy controls. (**a**) NMDS analysis in two groups. (**b**) PCoA analysis in two groups.

**Figure 3 nutrients-14-04582-f003:**
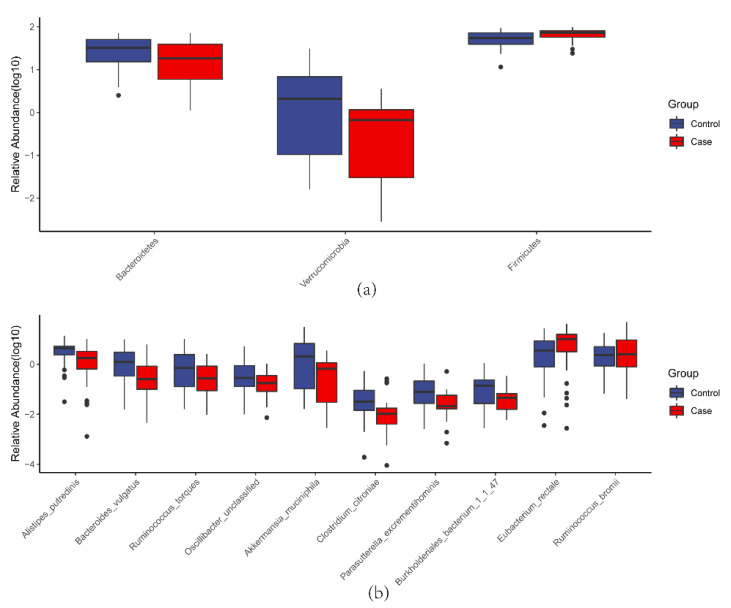
Boxplot of different microbial taxa. (**a**) Different phylum level taxa in two groups. (**b**) Different species level taxa in two groups.

**Figure 4 nutrients-14-04582-f004:**
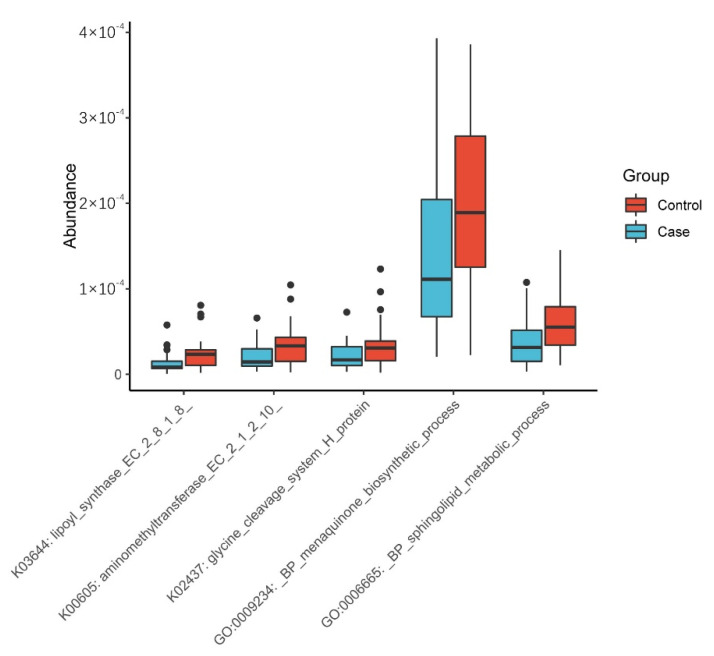
Boxplot of different function modules in two groups.

**Figure 5 nutrients-14-04582-f005:**
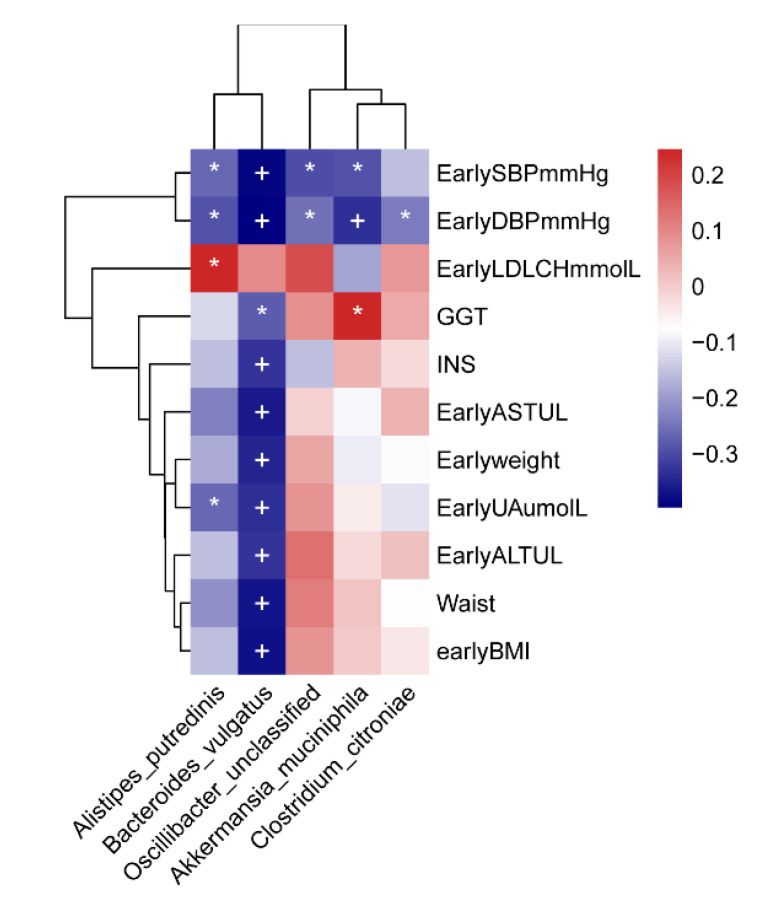
Heatmap of Spearman correlation between different species and clinical indices between case and control groups. Positive correlations are indicated in red text and negative correlations are indicated in blue text. (* *p* < 0.05; + *p* < 0.01). SBP, systolic blood pressure; DBP, diastolic blood pressure; LDLCH, low density lipoprotein cholesterol; GGT, glutamyltransferase; INS, insulin; AST, glutamic oxalacetic transaminase; UA, uric acid; ALT, glutamic pyruvic transaminase.

**Figure 6 nutrients-14-04582-f006:**
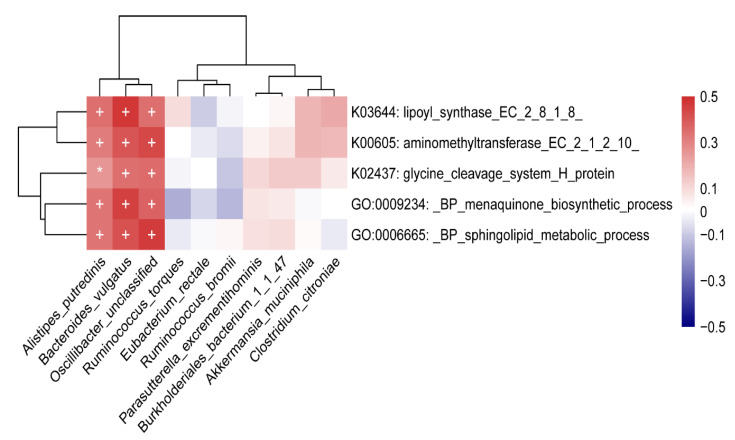
Heatmap of Spearman correlation between different species and function modules between case and control groups. Positive correlations are indicated in red text and negative correlations are indicated in blue text. (* *p* < 0.05; + *p* < 0.01).

**Table 1 nutrients-14-04582-t001:** Basic characteristics of subjects.

Variable	Control (N = 35)	PIH (N = 35)	*p*-Value
Age (year)	31.3 ± 3.9	31.6 ± 4.2	0.221
Gestational age (week)	12.70 ± 0.86	13.15 ± 3.26	0.428
Race			0.221
Han	33 (94.3%)	34 (97.1%)	
Others	2 (5.7%)	1 (2.9%)	
Occupation			0.954
Professionals	9 (25.7%)	8 (22.9%)	
Company employee	14 (40.0%)	15 (42.9%)	
Others	12 (34.3%)	12 (34.3%)	
Education			
Junior college and below	18 (51.4%)	14 (40.0%)	
Undergraduate and above	17 (48.6%)	21 (60.0%)	
Monthly income			0.334
10,000 and below	18 (51.4%)	22 (62.9%)	
10,000 and above	17 (48.6%)	13 (37.1%)	

**Table 2 nutrients-14-04582-t002:** Clinical characteristics of subjects.

Variable	Control (N = 35)	PIH (N = 35)	*p*-Value
Early waist (cm)	79.47 ± 7.01	82.27 ± 9.08	0.041
Early BMI (kg/m2)	21.80 ± 2.20	23.27 ± 3.56	0.004
Early SBP (mmHg)	114.77 ± 9.54	125.06 ± 7.82	<0.001
Early DBP (mmHg)	73.49 ± 8.46	81.74 ± 7.28	<0.001
HGB (g/L)	122.71 ± 10.51	127.61 ± 7.27	0.026
GLU (mmol/L)	4.52 ± 0.39	4.72 ± 0.47	0.062
ALB (g/L)	45.84 ± 2.26	44.99 ± 2.48	0.150
ALT (U/L)	17.57 ± 10.54	22.74 ± 16.28	0.123
AST (U/L)	18.46 ± 5.48	21.34 ± 8.15	0.093
CREA (umol/L)	43.23 ± 8.16	44.09 ± 7.44	0.640
UA (umol/L)	197.89 ± 38.93	230.62 ± 54.07	0.006
UREA (mmol/L)	2.50 ± 0.55	2.62 ± 0.78	0.497
TG (mmol/L)	1.42 ± 0.40	1.67 ± 0.65	0.057
TCHOL (mmol/L)	4.74 ± 0.86	4.60 ± 0.79	0.500
HDLCH (mmol/L)	1.89 ± 0.39	1.90 ± 0.44	0.960
LDLCH (mmol/L)	2.74 ± 0.78	2.55 ± 0.71	0.313
hsCRP (mg/L)	3.86 ± 4.20	4.95 ± 5.01	0.324
INS (mU/L)	19.19 ± 31.74	45.28 ± 60.05	0.020
GGT (U/L)	13.57 ± 5.79	22.33 ± 16.94	0.003

BMI, body mass index; SBP, systolic blood pressure; DBP, diastolic blood pressure; HGB, hemoglobin; ALB, albumin; ALT, glutamic pyruvic transaminase; AST, glutamic oxalacetic transaminase; CREA, creatinine; UA, uric acid; TG, triglyceride; TCHOL, total cholesterol; HDLCH, high density lipoprotein cholesterol; LDLCH, low density lipoprotein cholesterol; hsCRP, hypersensitive C-reactive protein; INS, insulin; GGT, glutamyltransferase.

## Data Availability

The data set supporting the results of this article has been deposited in the EMBL European Nucleotide Archive (ENA) under BioProject accession code PRJEB55996 [http://www.ebi.ac.uk/ena/data/view/PRJEB13870 (accessed on 26 October 2022)].

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
