# Peer review of "The Association between Gut Microbiome and Pregnancy-Induced Hypertension: A Nested Case–Control Study"

_nutrients, 2022, doi:10.3390/nu14214582_

Round 1
Reviewer 1 Report (Previous Reviewer 1)
The revised manuscript is much improved. I do though suggest in table 1 that when displaying age, one decimal point is enough. I have no further comments.
Author Response
Point 1: The revised manuscript is much improved. I do though suggest in table 1 that when displaying age, one decimal point is enough. I have no further comments.
Response 1: Thank you for your suggestion. I have revised the expression of age in table 1.

Reviewer 2 Report (New Reviewer)
Preeclampsia is a devastating hypertensive pregnancy disorder which pathophysiology is not fully understood, and there are no approved therapies of the disease. Recent studies demonstrate all the many ways how the microbiota contributes to health and disease. There is no definition of a healthy human microbiome and even less is known about maternal microbiome in normotensive and hypertensive pregnant women. It is not well understood whether the dysbiosis associated with preeclampsia is a new onset feature or whether pregnancy acts as a “second hit” to trigger the activation of a previously dormant microbiome. This field is yet to be studied and it is certainly interesting.
Author Response
Point 1: Preeclampsia is a devastating hypertensive pregnancy disorder which pathophysiology is not fully understood, and there are no approved therapies of the disease. Recent studies demonstrate all the many ways how the microbiota contributes to health and disease. There is no definition of a healthy human microbiome and even less is known about maternal microbiome in normotensive and hypertensive pregnant women. It is not well understood whether the dysbiosis associated with preeclampsia is a new onset feature or whether pregnancy acts as a “second hit” to trigger the activation of a previously dormant microbiome. This field is yet to be studied and it is certainly interesting.
Response 1: Thank you for your comment. I am glad you are interested in this manuscript.

This manuscript is a resubmission of an earlier submission. The following is a list of the peer review reports and author responses from that submission.
Round 1
Reviewer 1 Report
The manuscript is a case-control study comparing the gut microbiome in women with pregnancy induced hypertension compared to similar pregnant women. Several differences were reported:
Major
1) there is too much of a leap to saying that differences in the gut microbiome between the contrasted groups may cause PIH. Differences in the gut microbiome may relate to factors linked to hypertension in pregnancy without having any actual causal relationship to hypertension. There is no information provided to suggest that the microbiome is readily modifiable
2) the meaning of the five function modules is really not going to be understood by the vast majority of readers - yet, it is in the abstract as if it is fully understood
Minor
1) please state that the matched controls are healthy pregnant women not just healthy women
2) clarify whether the women with PIH could receive antihypertensive medication or not and, if so, discuss the potential confounding effect of this
Reviewer 2 Report
The main concern I have about this paper is that it is filled with the presumption that the bacterial changes drive the pregnancy induced hypertension, rather than the possibility that these women are more insulin resistant, with higher blood pressure, higher BMI - which equally could drive the alterations in microbiota. For example, there may be factors such as dietary fibre, dietary choices, medications, comorbidities that drive both changes, and they may not be related.
The entire paper needs to be reviewed taking this point into account.
The other problem is that combining preeclampsia and gestational hypertension may not be an appropriate thing to do - there may be differences in these two groups, and some sensitivity analysis of these two groups might be useful. Also, how the baseline characteristics differ between these two groups needs to be considered.
The changes in microbiota might actually exist well before pregnancy, due to insulin resistance, obesity, genetics, dietary issues or a range of other identified issues.
The background and discussion needs to be edited to make this point clear.
Round 2
Reviewer 2 Report
Not sure if there are issues in the review process, but my concerns for this study have not been addressed at all.
And the concerns of the other reviewer have also been addressed very superficially.